# How to evaluate word embeddings?
# On importance of data efficiency and simple supervised tasks

## Abstract

Maybe the single most important goal of representation learning is making subsequent learning faster. Surprisingly, this fact is not well reflected in the way embeddings are evaluated. In addition, recent practice in word embeddings points towards importance of learning specialized representations. We argue that focus of word representation evaluation should reflect those trends and shift towards evaluating what *useful* information is *easily* accessible. Specifically, we propose that evaluation should focus on data efficiency and simple supervised tasks, where the amount of available data is varied and scores of a *supervised* model are reported for each subset (as commonly done in transfer learning).

In order to illustrate significance of such analysis, a comprehensive evaluation of selected word embeddings is presented. Proposed approach yields a more complete picture and brings new insight into performance characteristics, for instance information about word similarity or analogy tends to be non–linearly encoded in the embedding space, which questions the cosine–based, unsupervised, evaluation methods. All results and analysis scripts are available online.

## 1 Introduction

Using word embeddings remains a standard practice in modern NLP systems, both in shallow and deep architectures [Goldberg, 2015]. By encoding information about words in a relatively simple algebraic structure [Arora et al., 2016] they enable fast transfer to the task of interest[1]. The importance of word representation learning has lead to developing multiple algorithms, but lack of principled evaluation hinders moving the field forward, which motivates developing more principled ways of evaluating word representations. Word embeddings are not only hard to evaluate, but also challenging to train. Recent practice shows that one often needs to tune algorithm, corpus and hyperparameters towards the target task [Lai et al., 2016, Sharp et al., 2016b], which challenges the promise of broad applicability of unsupervised pretraining.

Evaluation methods of word embeddings can be roughly divided into two groups: extrinsic and intrinsic [Schnabel and Labutov, 2015]. In the former approach embeddings are used in a downstream task (eg. POS tagging), while in the latter embeddings are tested directly for preserving syntactic of semantic relations. The most popular intrinsic task is Word Similarity (WS) which evaluates how well dot product between two vectors reproduce score assigned by human annotators. Intrinsic evaluations always assume a very specific model for recovering given property.

Despite popularity of word embeddings, there is no clear consensus what evaluation methods should be used, and both intrinsic and downstream evaluations are criticized [Tsvetkov et al., 2015a, Faruqui et al., 2016]. On top of that, different evaluation schemes usually lead to different rankings of embeddings [Schnabel and Labutov, 2015]. For instance, it has been shown [Baroni and Dinu, 2014] that neural-based word embeddings perform consistently better then count-based models and later, using WS and WA tasks, it was argued otherwise [Levy et al., 2015]. Recent research in evaluation methods focuses on representative or inter-

---

[1]Algebraic structure refers to the fact that words can be decomposed into a overcomplete basis, such that each word can be expressed as a sparse sum of base vectors

pretable set of tasks [Nayak et al., 2016, Köhn, 2015a], analysing intrinsic evaluation [Chiu et al., 2016, Faruqui et al., 2016], as well as proposing improvements to intrinsic evaluation [Avraham and Goldberg, 2016, Tsvetkov et al., 2015b].

In this paper we employ a transfer learning view, in which the main goal of representation learning is to make subsequent learning fast, i.e. use resulting word embeddings to maximize performance at the lowest sample complexity possible [Bengio et al., 2013, Glorot et al., 2011][2]. Surprisingly, researchers rarely report model (using given word representation) performance under varying (benchmark) dataset sizes and model classes[3], which is crucial for correct evaluation of transfer, especially given increasing importance of small data regime applications. Motivated by this, we propose an evaluation focused on data efficiency. To quantify precisely accessible information, we additionally propose focusing only on simple (supervised) tasks, as complex downstream tasks are challenging to interpret. In addition, we propose principled improvements to WS and WA tasks, which try to address some of the critiques both benchmarks have received in the literature [Faruqui et al., 2016], in authors' opinion mostly due to their purely unsupervised nature.

## 2 Proposal

Our main goal is to better align evaluation of word embeddings with their transfer application. Future performance is correlated with the amount of *easily* accessible and *useful* information. By *easily* accessible information, we mean information that model can quickly learn to use. *Useful* information is defined as one that correlates well with the final task.

First argument for data efficiency focused evaluation is the growing evidence that pretrained word embeddings provide little benefit under various settings, especially deep learning models [Zhang et al., 2015, Zhang and Wallace, 2015, Andreas and Klein]. We hypothesize that most of the improvements (in downstream tasks) reported in literature are caused by small size of the supervised dataset, which is reasonable from the trans-

---

[2]Alternative goals might include maximizing interpretability, or analysing unsupervised corpora.

[3]What is claimed here is that vast majority of papers doesn't take into consideration those factors. Nevertheless, there are notable exceptions [Andreas and Klein, Qu et al., 2015, Amir et al., 2017].

fer learning point of view. Therefore, measuring performance after seeing just a subset of the supervised dataset is crucial for comparing word embeddings. Another argument is the empiricial difference between how *easily* accessible is the information in various embeddings. As our experiment show, commonly used dense representations achieve different learning speeds. This effect should be even stronger for sparse representations, for which feature dimensions can have very strict semantic meaning [Faruqui and Dyer, 2015a]. An argument can be also made from theoretical point of view; it is easy to show that any injective (and thus not losing information) embedding preserves all information about corpora (see Appendix for details), i.e. having enough training data makes embeddings dispensable.

Second part of the proposal is to focus on simple supervised tasks to directly evaluate *useful* information content. In certain applications, like tagging, choosing the right, specialized, word embeddings is crucial for obtaining state of the art results [Sharp et al., 2016a, Lample et al., 2016]. We also confirm empirically that word embeddings trade off capacity between different information. In this work we pose hypothesis, that specialization of word embeddings can be best evaluated by checking what simple information is most *easily* recoverable. While word level classification problems (like noun classification) were proposed previously [Köhn, 2015b], here we also suggest including tests for recovery of relations between words (exemplified in experiments by *Similarity* and *Analogy* tasks) .

Importance of simple supervised tasks can be also seen in the light of algebraic structure that is encoded in word representation space. It has been observed in practice that word embeddings have *useful* information only in a small subspace [Sattigeri and Thiagarajan, 2016, Rothe and Schütze, 2016, Astudillo et al., 2015]. Thus, simple supervised tasks are closely aligned with the actual use of word embeddings and allow to quantify how quickly model can extract the most salient subspace (which leads to faster learning in general).

Our final remark is about diversity of models. Commonly used WS and WA datasets are solved by a *constant* models, i.e. model which does not learn from the data. We argue that such evaluation is not generally informative. If we are interested in how well our vector space helps solving a given

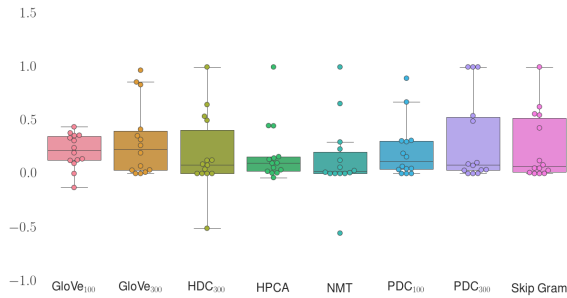

Figure 1: Vertical axis is normalized difference between score of the best performing non-linear model and the best performing linear model (the higher, the better non-linear model is). Each box plot represents a single tested embedding.

problem, we should in theory fit all possible models and pick the one that has the best generalization capabilities. While this is impractical, it illustrates that fixing one specific model gives answer to a different question, thus drawing general conclusions from it can be highly biased. A good rule of thumb might be to include representatives of typical model classes, or at least match the model with class of models we are interested in (which rarely will be *constant*), which concludes our guidelines for a correct evaluation.

We leave out details from the proposal how to order embeddings, as this is determined by the specific research question given evaluation should answer. A sensible default is to report AUC of learning curve for each *task*, and pick set of tasks that are most interesting to the researcher.

To summarize:

- Evaluation should focus on data efficiency (if transfer is the main goal of representation learning).

- Tasks should be supervised and simple.

- Unless focus is on specific application, evaluation should focus on a diverse set of models (including nonlinear and linear ones) and datasets (testing for various information content).

If we follow those guidelines, we truly approximate (for a given trained embedding) generalization error under distribution of tasks, dataset size and classifiers,

$$\mathbb{E}\left[\mathbf{L^t}(\mathrm{cl}(\mathbf{V_U}(\mathbf{X_m^t}), \mathbf{Y_m^t})(\mathbf{V_U}(\mathbf{x}), \mathbf{y})\right], \quad (1)$$

$\mathbf{V_U}$ denotes the embeddings trainedw on $\mathbf{U}$, $\mathbf{L^t}$ denotes task $\mathbf{t}$ loss and expectation is taken with respect to:

- $p(\mathrm{cl})$ – distribution of classifiers,

- $p(\mathbf{X_m^t}, \mathbf{Y_m^t})$ – distribution of training datasets, where we first sample task $t$ and then uniformly sample dataset $\mathbf{X}_m^t$ of size $m$.

- $\mathbf{x}, \mathbf{y}$ – i.i.d. examples following training data distribution.

Distribution over classifiers and tasks *should* be carefully tuned to researcher's needs, as we will argue soon. Further theoretical analysis is included in Appendix, and in the rest of the paper we present practical arguments for the proposed evaluation scheme.

## 3 Experiments

In this section we define specific metrics and tasks and perform exemplary evaluation of several pretrained embeddings in the advocated setting. Specifically, we try to empirically answer several questions, all geared towards providing experimental validation for the three main points of proposal:

- Do supervised versions of WA and WS benchmarks provide additional insights?

- How stable is the ranking of embeddings under changing dataset size?

- Are there embeddings that benefit from non-linear models?

The first question will aid understanding how useful are simple supervised tasks coupled with data efficiency. Second question shows that ranking of embeddings do change under transfer learning evaluation. Last question explores if any embeddings encode information in a "non-linear" fashion; while one of the main goals of representation learning is disentangling factors of variation, usually learned representations are entangled and dense, which poses interesting question how hard it is to extract useful patterns from them. In this paper we report only a subset of results with the most interesting conclusions, all results (along with code) are also made available online for further analysis.

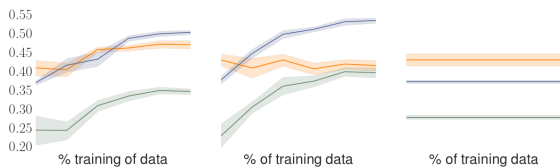

Figure 2: Accuracy reached by 3 different models optimizing word similarity on SimLex dataset. Rightmost plot deptics performance of the traditionally used *constant* model. While NMT has been reported to have strong performance on Sim-Lex (as shown on the rightmost plot), its relative gains diminish under supervised version of the benchmark (leftmost and central plot). Tested embeddings (ordered by performance on the rightmost plot): NMT, GloVe$_{300}$, HPCA$_{autoenc}$.

## 3.1 Datasets and models

Datasets are divided into 4 categories: *Similarity*, *Analogy*, *Sentence* and *Single word*. *Analogy* datasets are composed of quadruples (two pairs of words in a specific relation, for instance (king, queen, man, woman)). *Similarity* datasets are composed of pairs of words and assigned mean rank by human annotators. *Sentence* and *Single word* datasets have binary targets. In total our experimentation include 15 datasets:

- *Similarity*: SimLex999 [Hill et al., 2015], MEN [Bruni et al., 2014], WordSimilarity353 [Finkelstein et al., 2001] and Rare Words [Luong et al., 2013].

- *Analogy*: 4 categories from WordRep [Gao et al., 2014][4].

- *Sentence*: Stanford Sentiment Treebank [Socher et al., 2013] and News20 (3 binary datasets) [Tsvetkov et al., 2015a].

- *Single word*: Datasets constructed from lexicons collected in [Faruqui and Dyer, 2015b]: POS tagging (3 datasets for verb, noun and adjective), word sentiment (1 dataset), word color association (1 dataset) and WordNet synset membership (2 datasets).

Models for each datasets include both non-linear and linear variants. When model is non-

---

[4]We experimented with MSR and Google datasets and observed that models easily overfit if the train and test sets share the same *words* (not 3-tuples). WordRep dataset is a set of *pairs* which we split into disjoint sets.

linear, for robustness we include in search a fall-back to a simpler linear or *constant* model. Additionally, in the case of *Similarity* and *Analogy* we include commonly used *constant* models. Similarity between 2 vectors is approximated by their cosine similarity ($cos(\vec{v_1}, \vec{v_2})$). In the case of *Analogy* tasks embedding is evaluated for its ability to infer $4^{th}$ word out from the first three and we use the following well-known *constant* models: 3COSADD ($\arg\max_{\vec{v} \in \mathcal{V}} cos(\vec{v}, \vec{v_2} - \vec{v_1} + \vec{v_3})$) and 3COSMUL ($\arg\max_{\vec{v} \in \mathcal{V}} \frac{ccos(\vec{v}, \vec{v_3})ccos(\vec{v}, \vec{v_2})}{ccos(\vec{v}, \vec{v_1}) + \epsilon}$)[5]. For each task class we evaluate a different set of classifiers:

- *Similarity*: cosine similarity, Random Forest (RF), Support Vector Regression (SVR) with RBF kernel[6].

- *Analogy*: 3COSADD, 3COSMUL [Levy et al., 2015] and regression neural network (performing regression on the $4^{th}$ word given the rest of the quadruple, see Appendix for further information).

- *Sentence*: Logistic Regression, Support Vector Machine (SVM) with RBF kernel taking as input averaged embedding vector and Convolutional Neural Network (CNN) [Kim, 2014] taking as input concatenation of embedding vectors.

- *Single word*: RF, SVM (with RBF kernel), Naive Bayes, k-Nearest Neighbor Classifier and Logistic Regression.

## 3.2 Embeddings

Our objective was to cover representatives of embeddings emerging from both shallow and deeper architectures. Deep embeddings are harder to train, so for the scope of this paper we decided to include pretrained and publicly available vectors[7]. Setup includes following "shallow" pretrained embeddings: GloVe (100 and 300 dimensions) [Pennington et al., 2014], Hellinger PCA (HPCA) [Lebret and Collobert, 2014], PDC (100 and 300 dimensions) and HDC (300 dimensions) [Sun et al., 2015], Additionally following "deep" embeddings are evaluated: Neural Translation Machine (NMT,

---

[5]$ccos(\vec{v_1}, \vec{v_2}) = \frac{1 + cos(\vec{v_1}, \vec{v_2})}{2}$

[6]We also tried RankSVM [Lee and Lin, 2014], but it did not perform better than other models, while being very computationally intensive.

[7]Vocabularies were lowercased and intersected before performing experiments. Vectors were normalized to a unit length.

|  | rank start | rank end | auc rank |
|---|---|---|---|
| $GloVe_{100}$ | $2.09 \pm 1.76$ | $2.48 \pm 1.55$ | $2.17 \pm 1.77$ |
| $GloVe_{300}$ | $1.83 \pm 2.63$ | $1.91 \pm 2.54$ | $1.87 \pm 2.57$ |
| $HDC_{300}$ | $0.93 \pm 1.34$ | $1.06 \pm 1.22$ | $0.91 \pm 1.40$ |
| $HPCA_{autoenc}$ | $3.52 \pm 2.04$ | $3.68 \pm 1.67$ | $3.65 \pm 2.03$ |
| HPCA | $4.77 \pm 2.33$ | $4.75 \pm 2.23$ | $4.83 \pm 2.29$ |
| NMT | $3.90 \pm 2.24$ | $4.32 \pm 1.93$ | $4.28 \pm 2.34$ |
| $PDC_{300}$ | $0.64 \pm 0.89$ | $0.74 \pm 1.07$ | $0.65 \pm 1.10$ |
| morphoRNNLM | $4.13 \pm 2.34$ | $4.55 \pm 2.02$ | $4.26 \pm 2.29$ |

Table 1: First two column present ranks at the 30% and 100% splits of evaluation dataset averaged over all categories. Third column is rank computed by recommended default AUC of curve. As expected, when averaging over many tasks (of different dataset sizes), data efficiency is not changing final ordering (third column). On average rank increases as embeddings are becoming significantly different (as determined by ANOVA during rank computation).

activations of the deep model are extracted as word embeddings) [Hill et al., 2014], morphological embeddings (morph, which can learn morphological differences between words directly) [Luong et al., 2013] and HPCA variant trained using autoencoder architecture [Lebret and Collobert, 2015]. In some experiments we additionally include publicly available pretrained skip-gram embeddings on Google News corpora and skip-n-gram embeddings trained on Wikipedia corpora [Ling et al., 2015] (used commonly in syntax demanding tasks, like tagging).

### 3.3 Results

For each dataset we first randomly select test set and run evaluation for increasing sizes of training dataset, thus scores approximate generalization error after seeing increasing amounts of data. Splits are repeated 6 times in total to reduce noise. Thus, for each *task* results are 6 learning curves, with a score for each subset of data (see Fig. 2).

Ranks of embeddings at each point are calculated using a greedy sequential procedure, where we assign embeddings the same rank if their scores (each point on the curve is represented by 6 scores) are not significantly different, as tested using pairwise ANOVA test. All results are available online[8].

### 3.4 Learnable *Similarity* and *Analogy* tasks

Our first question was validating that adding learnable *Similarity* and *Analogy* tasks introduce any

---

[8]Results will be posted online upon publication.

|  | 3CosAdd | 3CosMul | NN |
|---|---|---|---|
| SimilarTo | 1% / 1% | 0% / 0% | 1% / 1% |
| InstanceOf | 1% / 1% | 1% / 1% | 22% / 26% |
| Antonym | 14% / 14% | 13% / 13% | 16% / 18% |
| DerivedFrom | 4% / 4% | 3% / 3% | 8% / 10% |

Table 2: Test accuracy at 30% and 100% training data achieved on different *Analogy* benchmarks using 3 different models (only NN is supervised), maximized over embeddings. Low *constant* model scores are similar to numbers reported in [Gao et al., 2014].

additional insights. Positive answer to this question motivates introduction of simple tasks with varying dataset size, ideally defined on single or pair of words.

For solving *Analogy* we implemented a shallow neural network. Interestingly, WordRep authors [Gao et al., 2014] reported low accuracies (often even below 5%) on most analogy questions and we were able to improve absolute score upon the tested subset on average by absolute 11% (see Tab. 2). Having learnable models for *Similarity* and *Analogy* datasets enables reusing many publicly available datasets in the new context. Also, we can robustly evaluate if given information about relation between two words is present in the embedding based on analogy questions. In the case of HASCONTEXT and INSTANCEOF datasets no embeddings can recover analogy answers using static models (achieved accuracy is below 3%), but actually some embeddings *do* have information about the relations. In both cases HDC consistently outperforms other embeddings reaching around 25% accuracy, see Fig. 4 and Tab. 2.

In the case of *Similarity* dataset the best performing model was Support Vector Regression, similarly as in *Analogy* datasets we also improve over the *constant* models. What is more, we can draw novel conclusions. Interesting example is NMT performance on SimLex. It was claimed in [Hill et al., 2014] that NMT embeddings are better at encoding *true* similarity between words, but SVR on Glove embeddings performs better after training on the whole dataset (i.e. at the end of learning curve), see Fig.2.

|  | rank start | rank end | auc rank |
|---|---|---|---|
| GloVe$_{100}$ | $1.80 \pm 2.04$ | $2.31 \pm 1.83$ | $1.86 \pm 2.07$ |
| GloVe$_{300}$ | $3.06 \pm 3.17$ | $3.14 \pm 2.99$ | $3.03 \pm 3.09$ |
| HDC$_{300}$ | $1.37 \pm 1.68$ | $1.37 \pm 1.40$ | $1.31 \pm 1.69$ |
| HPCA$_{autoenc}$ | $3.63 \pm 2.30$ | $3.80 \pm 1.98$ | $3.83 \pm 2.28$ |
| HPCA | $3.77 \pm 2.77$ | $3.60 \pm 2.58$ | $3.80 \pm 2.79$ |
| NMT | $3.74 \pm 2.05$ | $4.29 \pm 1.74$ | $4.23 \pm 2.30$ |
| PDC$_{300}$ | $0.60 \pm 0.77$ | $0.60 \pm 1.01$ | $0.51 \pm 0.95$ |
| morphoRNNLM | $3.00 \pm 2.38$ | $3.60 \pm 2.16$ | $3.17 \pm 2.38$ |

Table 3: First two column present ranks at the 30% and 100% splits of evaluation dataset averaged over all *tasks* for *Single word* datasets. Third column is rank computed by recommended default AUC of curve. Interestingly, information stored in *Single word* datasets is easily accessible even with small amount of data.

### 3.5 Rank stability under changing dataset size

Second question was how stable are the orderings under growing dataset. To this end we have measured rank at the beginning (30% of data) and end of training. Mean absolute value of change of ranking is approximately (averaged over all categories) 0.6 with standard deviation of 0.2. This means that usually an embeddings has a changed rank after training, which establishes usefulness of measuring data efficiency for the tested embeddings.

Interestingly, when averaged over many experiments, final *ordering* of embeddings tends to be similar, see Tab. 1 and Fig. 3. This is mainly because (tested) embeddings have different data efficiency properties for different tasks, i.e. none of embeddings is consistently more data efficient than others. On top of that standard deviation of both rank at the end and beginning is around 2.5, which further reinforces findings from [Schnabel and Labutov, 2015] that embeddings orderings are very *task* dependent.

Measuring data efficiency is crucial for a realistic (i.e. as close to application as possible) evaluation of representations. Besides a more accurate and practical ordering of embeddings, it also allows one to draw new conclusions, which is exemplified by differences between GloVe$_{100}$ and GloVe$_{300}$ (elaborated on in the next section). Another interesting point is that rank change after training on full dataset is relatively low for *Single word* datasets, which suggests that simple information about words like noun or verb is always quickly accessible to models, but more complicated information like relationships or similarities

between pair of words are not, see Tab. 3.

### 3.6 Linear vs non–linear models

Our last question was how stable is the ordering under changing model type. More specifically, are there embeddings especially fitted for use with linear models? Clearly some embeddings in fact are, see Fig. 1. This is an important empirical fact for practitioners, which motivates including such evaluation in experiments. In particular, it clearly shows that typically used evaluation does not answer the question "is there information about task X in the embedding Y" but only "is information about task X stored in embedding Y easily separable by a static (or linear) classifier".

An illustrative example is the difference in performance between two pretrained GloVe embeddings of different dimensionality (100 and 300). It has been shown previously that lower dimensional GloVe embeddings are better at syntactic tasks [Lai et al.], but our evaluation reveals more complicated picture, that GloVe objective might encourage some sort of nonlinear encoding. We can see that by significantly better rank of GloVe$_{100}$ at the beginning of learning of *Single word* datasets (mean rank 1.8), but lower at the end (mean rank 2.3), see Tab. 3. This is also visible when averaged over all categories, see Tab. 1.

### 3.7 Discussion

Performed eperiments show usefulness of the additional and more granular level of analysis enabled. Researcher can ask more precise questions, like "is it worth fitting syntax specific embeddings even when supervised dataset size is large?" (to which answer is positive based on our experiments) or "is HASINSTANCE relation encoded in the space?" (to which answer is also positive for some embeddings). Unfortunately, there is already a large volatility of final embeddings ordering when using standard evaluation, and our proposed scheme at times makes it even more challenging to decide which embeddings are optimal. This hints, that purely unsupervised large scale pretraining might not be suitable for NLP applications. Most importantly, evaluation should be more targeted, either at some specific application area, or at specific properties of representation.

One of the presented arguments for including supervised models for testing *information content* is algebraic interpretation of word embeddings [Arora et al., 2016]. The algebraic structure

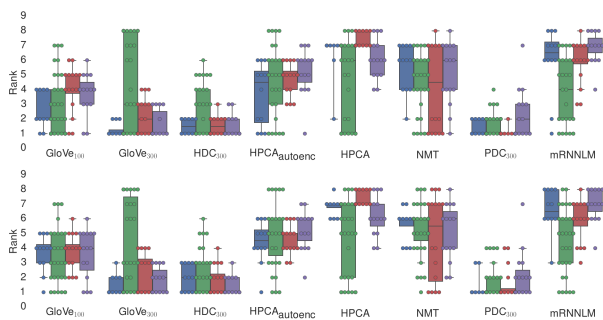

Figure 3: Rank at the beginning and end of learning for a subset of tested embedding (horizontal axis) and category (color). Task categories from left to right are: *Sentence* (blue), *Single word* (green), *Similarity* (red) and *Analogy* (purple).

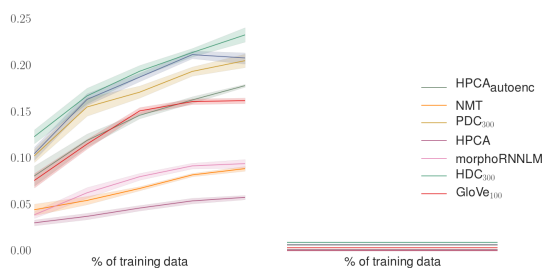

Figure 4: Accuracy reached by Neural Network regression model (left plot) and 3COSMUL (right plot) optimizing analogy (HASCONTEXT relation from WordRep benchmark [Gao et al., 2014]).

present in the representation space enables one to decompose word embedding space into a set of *concepts* (so each word vector can be well approximated by a sum of few *concepts*)[9]. Theoretically, tasks defined on single words should test for existence of such *concepts*, but in our case including (supervised) *Analogy* tasks was very useful, as those tasks are still very challenging for current embeddings. For *Analogy* tasks (see Fig. 4) achieved accuracy scores are below 25%, whereas in the case of *Single word* average accuracy is around 80% (and fitting classifier adds on average only 2%). These higher order (or subspace) focused tasks are also well aligned with the application of word embeddings, because empirically models tend to focus on a small subspace in the vector space.

---

[9]This decomposition can be obtained using standard methods like k-SVD.

## 4 Conclusions

As exemplified by experiments, proposed evaluation reveals differences between embeddings along usually overlooked dimensions: data efficiency, non-linearity of downstream model and simple supervised tasks (including recovery of higher order relations between words). Interesting new conclusions can be reached, including differences between different size GloVe embeddings or performance of non-linear models on similarity benchmarks.

Additionally, obtained results reinforce conclusions from other published studies that there are no universally good embeddings and finding such might not be achievable, or a well posed problem. One should take great care when designing evaluation and specify what is the main focus. For instance, if the main goal of the word embeddings is to be useful in transfer, one should include advocated data efficiency metrics. New word embedding algorithms are moving away from typical pretraining scheme, with increasing focus on specialized word embeddings and applications under very limited dataset size, where fast learning is crucial. We hope that proposed evaluation methodology will help advance research in these scenarios.

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

# A  Theoretical analysis

Let us first try to answer the question what is the *information about the task* and how can it be measured given some data representation. For simplicity let us assume that task is a binary classification, but the same reasoning applies to multiclass, multilabel, regressions etc. A quite natural, machine learning perspective is to define information stored as a Bayes risk of optimal model trained to perform this task. Obviously, raw representation already has some non-negative Bayes risk, which cannot be reduced during any embedding. Actually, it is quite easy to show, that nearly every embedding preserves all the information contained in the source representation.

**Observation 1.** *Every injective embedding preserves all the information about the task.*

*Proof.* Let us assume that Bayesian optimal classifier (the one obtaining the Bayes risk $\mathcal{R}_{\mathcal{X}}$) on the input space $\mathcal{X}$ be called $o$. Furthermore, let our embedding (learned in arbitrary manner, supervised or not) be a function $\mathcal{E} : \mathcal{X} \to \mathcal{X}'$. According to assumptions, $\mathcal{E}$ is injective, thus for every $x \in \mathcal{X}$ there exists unique $x' \in \mathcal{X}'$ such that $\mathcal{E}(x) = x'$. Let us call the corresponding inverse assignment $\mathcal{E}^{-1}$ (defined only on the image of $\mathcal{E}$). Consequently we can define classifier on $\mathcal{E}(X) \subset \mathcal{X}'$ through $o'(x') = o(\mathcal{E}^{-1}(x'))$. It is now easy to show, that Bayes risks of these two models are exactly the same

$$
\begin{aligned}
\mathcal{R}_{\mathcal{X}} &= \int \sum_y \ell(o(x), y) p(y|x) p(x) dx \\
&= \int \sum_y \ell(o(\mathcal{E}^{-1}(\mathcal{E}(x))), y) p(y|x) p(x) dx \\
&= \int \sum_y \ell(o'(\mathcal{E}(x), y) p(y|x) p(x) dx \\
&= \mathcal{R}_{\mathcal{X}'}.
\end{aligned}
$$

$\square$

This remains an open question how frequent in general are injective embeddings. If one considers continuous spaces as $\mathcal{X}$ then this is extremely small class of functions (especially if $\dim(\mathcal{X}') \leq \dim(\mathcal{X})$). However in case of natural language processing (and many other fields), the input space is actually discrete or even finite. In such case, non-injective embeddings are rare phenomenon. In particular, for any finite set, probability of selecting at random linear projection which gives non-injective embedding is zero.

The above reasoning is in some sense trivial, yet still worth underlying, as it gives an important notion of what should be measured when evaluating embeddings. Even though Bayes risk is the same for both spaces, the complexity of inferring $o'$ can be completely different from complexity of inferring $o$. We argue, that this is a crucial element - to measure how hard is to learn $o'$ (or any reasonable approximation). There are two basic dimensions of such analysis:

- check how complex the set of hypotheses $\mathcal{H} \ni o'$ needs to be in order to be able to find it using given data,

- verify how well one can approximate $o'$ as a function of growing training size. In other words - how fast an estimator of $o'$ converges.

Thus, to really distinguish various embeddings, we should rather ask what is the best achievable performance under limited amount of data or under constrained class of models, which is theoretical argument for data efficiency oriented evaluation.

## B Regression neural network for word analogy task

Let $D$ be the dimension of a given word embedding. We assume that all embedded words have euclidean norm equal to one – this guarantees that the scalar product of two embedded words is also their cosine similarity. The word analogy task is defined in the following way: given (embedded) words $v_1$, $v_2$ and $v_3$, predict word $v_4$ that satisfies analogy "$v_1$ is related to $v_2$ as $v_3$ is related to $v_4$". Our estimator of $v_4$ is defined as:

$$
\widehat{v_4} = \frac{-W_1 v_1 + W_2 v_2 + W_3 v_3 + b}{\|-W_1 v_1 + W_2 v_2 + W_3 v_3 + b\|_2}
$$

where the model trainable parameters are:

- $W_1, W_2$ and $W_3 - D \times D$ matrices initialized with identities,

- $b - D$-dimensional vector initialized with zeros,

and the cost is defined as:

$$
-\sum_j \langle v_4^j, \widehat{v_4^j} \rangle.
$$

The model was trained with gradient descent optimization on minibatches. Hyperparameters: learning rate, number of epochs, optimizer, batch size and (boolean) fallback to *constant* model were chosen using cross-validation. The actual prediction has two steps:

- calculate $\widehat{v_4}$,

- choose (embedded) word $v$ that minimizes $\langle v, \widehat{v_4} \rangle$.

Observe that this model is initialized in such a way, that it is equivalent to 3COSADD – the idea is to check, if applying trainable *affine* transformations to input vectors would boost 3COSADD performance. It should also be noted that this approach turned out to be very computationally intensive.

