# Peer review of "How to evaluate word embeddings? On importance of data efficiency and simple supervised tasks"

_ACL 2017 — decision unknown_

[Official Review · Reviewer 1 · rating 2 · confidence 3]
soundness 3 · originality 3 · clarity 1 · impact 3 · substance 3 · appropriateness 5 · meaningful comparison 3 · presentation format Poster

This paper proposes a framework for evaluation of word embeddings based on data
efficiency and simple supervised tasks. The main motivation is that word
embeddings are generally used in a transfer learning setting, where evaluation
is done based on how faster is to train a target model. The approach uses a set
of simple tasks evaluated in a supervised fashion, including common benchmarks
such as word similarity and word analogy. Experiments on a broad set of
embeddings show that ranks tend to be task-specific and change according to the
amount of training data used.

Strengths

- The transfer learning / data efficiency motivation is an interesting one, as
it directly relates to the idea of using embeddings as a simple
"semi-supervised" approach.

Weaknesses

- A good evaluation approach would be one that propagates to end tasks.
Specifically, if the approach gives some rank R for a set of embeddings, I
would like it to follow the same rank for an end task like text classification,
parsing or machine translation. However, the approach is not assessed in this
way so it is difficult to trust the technique is actually more useful than what
is traditionally done.
- The discussion about injective embeddings seems completely out-of-topic and
does not seem to add to the paper's understanding.
- The experimental section is very confusing. Section 3.7 points out that the
analysis results in answers to questions as "is it worth fitting syntax
specific embeddings even when supervised datset is large?" but I fail to
understand where in the evaluation the conclusion was made.
- Still in Section 3.7, the manuscript says "This hints, that purely
unsupervised large scale pretraining might not be suitable for NLP
applications". This is a very bold assumption and I again fail to understand
how this can be concluded from the proposed evaluation approach.
- All embeddings were obtained as off-the-shelf pretrained ones so there is no
control over which corpora they were trained on. This limits the validity of
the evaluation shown in the paper.
- The manuscript needs proofreading, especially in terms of citing figures in
the right places (why Figure 1, which is on page 3, is only cited in page 6?).

General Discussion

I think the paper starts with a very interesting motivation but it does not
properly evaluate if their approach is good or not. As mentioned above, for any
intrinsic evaluation approach I expect to see some study if the conclusions
propagate to end tasks and this is not done in the paper. The lack of clarity
and proofreading in the manuscript also hinders the understanding. In the
future, I think the paper would vastly benefit from some extrinsic studies and
a more controlled experimental setting (using the same corpora to train all
embeddings, for instance). But in the current state I do not think it is a good
addition to the conference.

[Official Review · Reviewer 2 · rating 3 · confidence 4]
soundness 3 · originality 3 · clarity 3 · impact 3 · substance 3 · appropriateness 5 · meaningful comparison 3 · presentation format Oral Presentation

- Strengths:

This paper proposed an interesting and important metric for evaluating the
quality of word embeddings, which is the "data efficiency" when it is used in
other supervised tasks.

Another interesting point in the paper is that the authors separated out three
questions: 1) whether supervised task offers more insights to evaluate
embedding quality; 2) How stable is the ranking vs labeled data set size; 3)
The benefit to linear vs non-linear models.

Overall, the authors presented comprehensive experiments to answer those
questions, and the results see quite interesting to know for the research
community.

- Weaknesses:

The overall result is not very useful for ML practioners in this field, because
it merely confirms what has been known or suspected, i.e. it depends on the
task at hand, the labeled data set size, the type of the model, etc. So, the
result in this paper is not very actionable. The reviewer noted that this
comprehensive analysis deepens the understanding of this topic.

- General Discussion:

The paper's presentation can be improved. Specifically: 

1) The order of the figures/tables in the paper should match the order they are
mentioned in the papers. Right now their order seems quite random.

2) Several typos (L250, 579, etc). Please use a spell checker.

3) Equation 1 is not very useful, and its exposition looks strange. It can be
removed, and leave just the text explanations.

4) L164 mentions the "Appendix", but it is not available in the paper.

5) Missing citation for the public skip-gram data set in L425.

6) The claim in L591-593 is too strong. It must be explained more clearly, i.e.
when it is useful and when it is not.

7) The observation in L642-645 is very interesting and important. It will be
good to follow up on this and provide concrete evidence or example from some
embedding. Some visualization may help too.

8) In L672 should provide examples of such "specialized word embeddings" and
how they are different than the general purpose embedding.

9) Figuer 3 is too small to read.